# Stable and Diverse Strategy Learning via Diffusion-Based Co-Evolution in StarCraft II Combat

## Abstract

Effective learning algorithms for agents in multi-agent environments remain a central challenge due to inter-agent dependencies during both training and evaluation. This challenge is amplified by the curl of the fitness landscape, which induces cyclic trajectories – especially in competitive settings. To address these challenges, we propose Diffusion Co-evolution, an evolutionary learning framework inspired by diffusion processes. Our method facilitates robust opponent matching by enabling broad exploration across a diverse agent population. It leverages quality diversity to identify multiple high-performing strategies, even in environments with cyclic learning dynamics. Experiments in a StarCraft II combat environment demonstrate Diffusion Co-evolution's superior stability and strategic diversity compared to conventional co-evolutionary baselines.

## 1 Introduction

Real-time decision-making in multi-agent systems is a central challenge in artificial intelligence, with applications ranging from autonomous driving and robotics to resource management and strategic decision support. These environments often involve multiple agents with competing or cooperative objectives, requiring adaptive strategies under uncertainty and time constraints. Competitive games provide an ideal testbed for studying such dynamics, as they capture the complexity of real-world interactions within a controlled experimental setting (Balduzzi et al., 2018; 2019).

In these contexts, agents can often approach optimal strategies through self-play (Heinrich et al., 2015), with convergence typically occurring toward a Nash equilibrium when all participants behave rationally. However, when the equilibrium does not correspond to a pure strategy, achieving convergence becomes difficult in practice. In such cases, agents must diversify their behaviors and adopt mixed strategies to approximate equilibrium play.

This challenge is exemplified in StarCraft II, a representative multi-agent competitive game that demands both rapid real-time decision-making and long-term strategic planning (Samvelyan et al., 2019; Vinyals et al., 2019; Ellis et al., 2023). Population-based optimization methods, such as evolutionary algorithms (Salimans et al., 2017; Hansen & Ostermeier, 2001; Hansen, 2016), provide notable advantages in this setting. By evaluating multiple individuals in each generation, these algorithms maintain population diversity and are generally less prone to premature convergence at local optima compared to gradient-based methods.

Among these approaches, diffusion evolution (Zhang et al., 2025) is particularly promising. Inspired by the diffusion process (Sohl-Dickstein et al., 2015; Ho et al., 2020; Song et al., 2021), it enhances population diversity and employs adaptive step-size control, thereby improving robustness against learning instability. Furthermore, due to its similarity to diffusion dynamics, the algorithm is capable of exploring multiple promising strategies or behavioral modes rather than converging narrowly on a single solution.

The remainder of this paper is organized as follows. Section 2 reviews related work, presents background information, and introduces the problem formulation. Section 3 details our proposed method, diffusion co-evolution. Section 4 presents experimental results in the StarCraft II environ-

ment, demonstrating both the performance and strategic diversity of our approach. Finally, Section 5 summarizes our contributions and highlights directions for future research.

## 2 BACKGROUND

### 2.1 DIFFUSION EVOLUTION

Diffusion Evolution (Zhang et al., 2025) is an optimization method inspired by the diffusion process (Ho et al., 2020). Diffusion-based models have been successfully applied to image and video generation tasks, and they are increasingly regarded as promising tools for agent training and planning. Building on the denoising process of DDIM (Song et al., 2021), defined as $p(\boldsymbol{x}_0|\boldsymbol{x}_t)$, diffusion evolution generates a population of candidate solutions $\boldsymbol{x}_0$ from an initial population of noise $\boldsymbol{x}_T$. This is achieved by progressively estimating the distribution of optimal solutions $p(\hat{\boldsymbol{x}}_0)$ from the current population $p(\boldsymbol{x}_t)$, as formalized in Equations 1–4. Here, $\boldsymbol{w} \sim \mathcal{N}(0, I)$ represents Gaussian noise, and $\sigma_t \in [0, 1]$ controls the magnitude of injected noise for exploration.

$$p(\boldsymbol{x}_0 = \boldsymbol{x}|\boldsymbol{x}_t) = \frac{p(\boldsymbol{x}_t|\boldsymbol{x}_0 = \boldsymbol{x})p(\boldsymbol{x}_0 = \boldsymbol{x})}{p(\boldsymbol{x}_t)} \tag{1}$$

$$= \frac{\mathcal{N}(\boldsymbol{x}_t; \sqrt{\alpha_t}\boldsymbol{x}_0, (1 - \alpha_t)I)g(f(\boldsymbol{x}))}{Z} \tag{2}$$

$$p(\hat{\boldsymbol{x}}_0) \propto \sum_{v \in \boldsymbol{x}_t} p(\boldsymbol{x}_0|\boldsymbol{x}_t = v)p(\boldsymbol{x}_t = v) \tag{3}$$

$$\boldsymbol{x}_{t-1} = \sqrt{\alpha_{t-1}}\hat{\boldsymbol{x}}_0 + \sqrt{1 - \alpha_{t-1} - \sigma_t^2}\hat{\epsilon} + \sigma_t\boldsymbol{w} \tag{4}$$

Diffusion evolution can be viewed as an evolutionary algorithm, since it iteratively updates a population of $N$ individuals over $T$ optimization steps. CMA-ES (Hansen & Ostermeier, 2001; Hansen, 2016) is a representative evolutionary strategy, which models a population mainly with a mean vector $\boldsymbol{\mu}$ and a covariance matrix $\boldsymbol{\Sigma}$, and updates this distribution by sampling individuals and evaluating their fitness. Unlike CMA-ES, diffusion evolution does not track aggregate statistics such as a mean or variance. Instead, each individual retains its own data point, and individuals interact based on their relative fitness values. This mechanism enables the population to self-organize and maintain multiple high-fitness solutions, thereby preserving diversity while optimizing performance.

### 2.2 PROBLEM FORMULATION

In this study, we adopt the StarCraft II environment, which features a stochastic reward structure and a vast action space, making it considerably more complex than deterministic environments with discrete action spaces. We model the StarCraft II environment as a win-rate function

$$f_{agent} : \Theta \times \Theta \times \mathcal{S} \to [0, 1], \quad f_{agent}(\theta_1, \theta_2|S) = p \tag{5}$$

where $\mathcal{S}$ denotes the state space of the game, $\theta_1, \theta_2 \in \Theta$ represent the parameters of the two agents, and $p$ is the win-rate of the first player. To focus on the impact of macro-management strategies and planning, we limit the scope of our analysis by ignoring micro-management factors. Specifically, we assume that both agents have equivalent micro-management skills, which remain fixed throughout training. Within this setting, we study the unit composition optimization problem in StarCraft II. Each player allocates resources $x_1, x_2$ across different unit types, after which the outcome is determined by simulating combat between the two unit compositions.

$$f_1, f_2 : \mathcal{X}_1 \times \mathcal{X}_2 \times \mathcal{S} \to [0, 1], \quad f_1(x_1, x_2|S) = 1 - f_2(x_1, x_2|S) \tag{6}$$

for all $x_1 \in \mathcal{X}_1, x_2 \in \mathcal{X}_2$, where $f_1$ and $f_2$ denote the win-rate functions of the first and second players, respectively. Our objective is to identify optimal strategies $x_1$ for the first player by maximizing $f_1(x_1, x_2 \mid S)$, under the assumption that the adversary selects $x_2$. Rather than focusing on a

single optimal point, we aim to discover the set of non-dominated strategies that preserve population diversity:

$$X_1 = \{x_1 : |\mathbb{E}_{x_2 \sim p(x_2)}[f_1(x_1, x_2)] - r^*| \leq \epsilon\} \tag{7}$$

where $r^* = \max_{x_1} \mathbb{E}_{x_2 \sim p(x_2)}[f_1(x_1, x_2)]$ and $\epsilon$ is a small tolerance. In simpler terms, this setup models StarCraft II as a resource allocation game: players distribute resources across unit types, and the goal is to find not just the best composition but a diverse set of strong strategies that remain effective against different opponents.

## 3 DIFFUSION CO-EVOLUTION

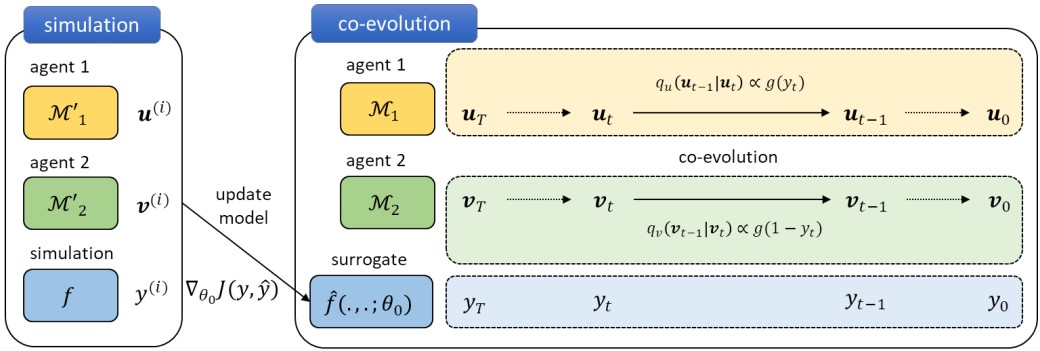

Figure 1: Overview of the proposed method.

Figure 1 and Algorithm 1 illustrate the overall process of the proposed method. The procedure begins by sampling strategies and evaluating their performance through simulation. Each simulation result is stored as a tuple $(\boldsymbol{u}, \boldsymbol{v}, y)$ in a dataset $\mathcal{D}$, which is then used to train a surrogate simulator $\hat{f}$. This surrogate, implemented as a neural network, approximates the original simulation and enables faster evaluations, making it suitable for real-time decision-making.

Next, the co-evolution process evaluates and updates agents using the surrogate model, with diffusion evolution as the underlying optimization algorithm. Diffusion evolution generates candidate strategies $\boldsymbol{x}_0$ according to

$$p(\boldsymbol{x}_0 = \boldsymbol{x}) = g[f(\boldsymbol{x})] \tag{8}$$

where $f$ denotes the fitness function and $g$ is an order-preserving rescaling function. In this work, we adopt the Boltzmann distribution as the rescaling function,

$$g(\boldsymbol{x}) \propto \exp(-\frac{\boldsymbol{x}}{c_T}) \tag{9}$$

which enables control over the balance between exploration and exploitation through the temperature parameter $c_T$. Following Zhang et al. (2025), we employ cosine scheduling (Nichol & Dhariwal, 2021),

$$\alpha_t = \frac{\cos(\pi t/T) + 1}{2} \tag{10}$$

for the diffusion schedule, and set the noise parameter to $\sigma_m = 1.0$.

## 4 EXPERIMENT AND RESULT

We first set up the StarCraft II Multi-agent Combat Environment as a testbed. The simulation code was implemented using the PySC2 framework (Vinyals et al., 2017) and executed with the SCII binary v4.10 on Linux. By mapping resource assignments to unit compositions via Algorithm 3, the task of maximizing win-rate through combat unit composition was reformulated as the problem of optimizing resource allocation, following the approach of Lee et al. (2021). In this formulation, the fitness function is

$$f : \Delta^D \times \Delta^D \to \mathbb{R} \tag{11}$$

---

**Algorithm 1** Diffusion Co-evolution

---

**Require:** Population size $N$, Parameter dimension $D$, Fitness function $f$, Density mapping function $g : x \mapsto \exp(-x/c_T)$, Diffusion step $T$, Diffusion schedule $\alpha$, and Noise schedule $\sigma_t = \sigma_m \sqrt{\frac{1-\alpha_{t-1}}{1-\alpha_t}} \sqrt{1 - \frac{\alpha_t}{\alpha_{t-1}}}$

1: $\mathcal{D} = \{\}; \theta_0.\text{INITIALIZE}()$
2: **for** $i \leftarrow 1$ **to** $N_{sim}$ **do**          $\triangleright$ Collect data from the simulation.
3:     $\boldsymbol{u} \leftarrow \mathcal{M}_1'.\text{SAMPLE}()$
4:     $\boldsymbol{v} \leftarrow \mathcal{M}_2'.\text{SAMPLE}()$
5:     $y \leftarrow f(\boldsymbol{u}, \boldsymbol{v})$
6:     $\mathcal{D} \leftarrow \mathcal{D} \cup \{(\boldsymbol{u}, \boldsymbol{v}, y)\}$
7: **end for**

8: **for** $i \leftarrow 1$ **to** $N_{train}$ **do**          $\triangleright$ Train a surrogate model
9:     $(\boldsymbol{u}, \boldsymbol{v}, y) \leftarrow \mathcal{D}.\text{SAMPLE}()$
10:    $\hat{y} \leftarrow \hat{f}(\boldsymbol{u}, \boldsymbol{v}; \theta_0)$
11:    $\theta_0.\text{UPDATE}(\nabla_{\theta_0} J(y, \hat{y}))$
12: **end for**

13: $\boldsymbol{U}_T = \langle \boldsymbol{u}_T^{(1)}, \boldsymbol{u}_T^{(2)}, \cdots, \boldsymbol{u}_T^{(N)} \rangle \leftarrow \mathcal{N}(\boldsymbol{0}, \mathbf{I}^{N \times D})$      $\triangleright$ Initialize two strategies
14: $\boldsymbol{V}_T = \langle \boldsymbol{v}_T^{(1)}, \boldsymbol{v}_T^{(2)}, \cdots, \boldsymbol{v}_T^{(N)} \rangle \leftarrow \mathcal{N}(\boldsymbol{0}, \mathbf{I}^{N \times D})$
15: **for** $t \leftarrow T$ **to** $2$ **do**
16:    $G_{ij} \leftarrow g[\hat{f}(\boldsymbol{u}_t^{(i)}, \boldsymbol{v}_t^{(j)}; \theta_0)], \forall i, j \in \{1, \cdots, N\}$      $\triangleright$ Evaluate fitness values
17:    **for** $k \in \{1, 2\}$ **do**
18:      **if** $k = 1$ **then**
19:        $\langle \boldsymbol{x}_t^{(1)}, \boldsymbol{x}_t^{(2)}, \cdots, \boldsymbol{x}_t^{(N)} \rangle \leftarrow \boldsymbol{U}_t$
20:        $Q_i \leftarrow \sum_{j=1}^N G_{ij}, \forall i \in \{1, \cdots, N\}$
21:      **else**
22:        $\langle \boldsymbol{x}_t^{(1)}, \boldsymbol{x}_t^{(2)}, \cdots, \boldsymbol{x}_t^{(N)} \rangle \leftarrow \boldsymbol{V}_t$
23:        $Q_j \leftarrow \sum_{i=1}^N G_{ij}, \forall j \in \{1, \cdots, N\}$
24:      **end if**
25:      **for** $i \leftarrow 1$ **to** $N$ **do**          $\triangleright$ Diffusion Evolution
26:        $Z \leftarrow \sum_{j=1}^N Q_j \mathcal{N}(\boldsymbol{x}_t^{(i)}; \sqrt{\alpha_t} \boldsymbol{x}_t^{(j)}, 1 - \alpha_t)$
27:        $\hat{\boldsymbol{x}}_0 \leftarrow \frac{1}{Z} \sum_{j=1}^N Q_j \mathcal{N}(\boldsymbol{x}_t^{(i)}; \sqrt{\alpha_t} \boldsymbol{x}_t^{(j)}, 1 - \alpha_t) \boldsymbol{x}_t^{(j)}$
28:        $\boldsymbol{w} \leftarrow \mathcal{N}(0, I^D)$
29:        $\hat{\epsilon} \leftarrow \frac{\boldsymbol{x}_t^{(i)} - \sqrt{\alpha_{t-1}} \hat{\boldsymbol{x}}_0}{\sqrt{1-\alpha_t}}$
30:        $\boldsymbol{x}_{t-1}^{(i)} \leftarrow \sqrt{\alpha_{t-1}} \hat{\boldsymbol{x}}_0 + \sqrt{1 - \alpha_{t-1} - \sigma_t^2} \hat{\epsilon} + \sigma_t \boldsymbol{w}$
31:      **end for**
32:      **if** $k = 1$ **then**
33:        $\boldsymbol{U}_{t-1} \leftarrow \langle \boldsymbol{x}_{t-1}^{(1)}, \boldsymbol{x}_{t-1}^{(2)}, \cdots, \boldsymbol{x}_{t-1}^{(N)} \rangle$
34:      **else**
35:        $\boldsymbol{V}_{t-1} \leftarrow \langle \boldsymbol{x}_{t-1}^{(1)}, \boldsymbol{x}_{t-1}^{(2)}, \cdots, \boldsymbol{x}_{t-1}^{(N)} \rangle$
36:      **end if**
37:    **end for**
38: **end for**

---

where $\Delta^D$ denotes a $D$-dimensional simplex. To construct scenarios with multiple optimal strategies, we varied the maximum available resources $\boldsymbol{b}$ for each player. For each scenario, we randomly sampled $N = 40$ strategies $\underset{iid}{\sim}$ Dirichlet($\boldsymbol{1}$) distribution and evaluated their win-rates using $r = 10$ simulations, as described in Algorithm 2.

Our experiments identified two scenarios, $b = \langle 5000, 2500, 75 \rangle$ (5000 minerals, 2500 gases, and 75 food) and $b = \langle 5000, 4000, 75 \rangle$, that yielded multiple non-dominated pure strategies (Figure 2). In the first scenario ($b = \langle 5000, 2500, 75 \rangle$, Figure 2a), the best strategy achieves a higher win-rate against the second-best strategy, which in turn outperforms the third-best strategy. However, the

---

**Algorithm 2** Visualizing game matrix

---

    **Input**: Strategy generator $\mathcal{M}'$, Fitness function $f'$
    **Output**: Value matrix $\boldsymbol{B}$
1: **procedure** ESTIMATION($f', \mathcal{M}'$)
2:     $\boldsymbol{U}' \leftarrow \mathcal{M}'$.GENERATE()
3:     **for** $i \leftarrow 1$ **to** $N$ **do**
4:         **for** $j \leftarrow 1$ **to** $N$ **do**
5:             $a_{ij} \leftarrow 0$
6:             **for** $t \leftarrow 1$ **to** $r$ **do**         ▷ Simulations repeated for $r$-times
7:                 $a_{ij} \leftarrow a_{ij} + f'(\boldsymbol{u}'^{(i)}, \boldsymbol{u}'^{(j)})$     ▷ Evaluate each element of a matrix
8:             **end for**
9:             $a_{ij} \leftarrow a_{ij}/r$
10:        **end for**
11:    **end for**
12:    $\mathbf{B} \leftarrow (\mathbf{A} - \mathbf{A}^\top)/2$         ▷ Guarantees antisymmetry $\mathbf{B}^\top = -\mathbf{B}$
13:    **return** B
14: **end procedure**

---

Table 1: Unit types and unit costs.

| Unit Type | Mineral | Gas | Food |
|-----------|---------|-----|------|
| Zealot | 100 | 0 | 2 |
| Sentry | 50 | 100 | 2 |
| Stalker | 125 | 50 | 2 |
| Adept | 100 | 25 | 2 |
| Immortal | 250 | 100 | 4 |
| Colossus | 300 | 200 | 6 |

third-best strategy performs better against the first, illustrating a cyclic dominance relationship. Similarly, in the second scenario ($b = \langle 5000, 4000, 75 \rangle$, Figure 2b), the three best-performing strategies counter one another. In contrast, under $b = \langle 3000, 2500, 75 \rangle$ (Figure 2c), such cyclical interactions do not emerge.

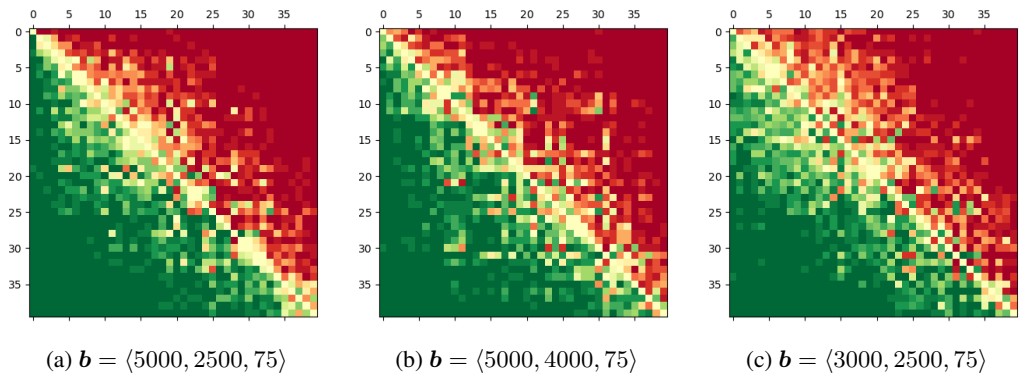

(a) $\boldsymbol{b} = \langle 5000, 2500, 75 \rangle$          (b) $\boldsymbol{b} = \langle 5000, 4000, 75 \rangle$          (c) $\boldsymbol{b} = \langle 3000, 2500, 75 \rangle$

Figure 2: Win-rates of $40$ randomly sampled unit compositions in specific scenarios, sorted by average win-rate. Each cell indicates the outcome of the row player against the column player: green denotes that the row player's composition outperforms the column player's, while red indicates the opposite.

For the two selected scenarios, we evaluated the performance of the proposed method against baseline algorithms using a population size of $N = 16$ and $T = 200$ generations. Since the search space lies in $\mathbb{R}^D \times \mathbb{R}^D$, we applied a softmax transformation to ensure the constraint $\|\boldsymbol{x}\|_1 = 1$ before fitness evaluation.

---

**Algorithm 3** Map resource assignment ratio to unit-combination.

---

**Require:** resource assignment ratio $\boldsymbol{z} \in \mathbb{R}_+^D$, $||\boldsymbol{z}||_1 = 1$, cost matrix $\boldsymbol{A} \in \mathbb{R}^{D \times 3}$ ($D = 6$, see Table 1), resource limit $\boldsymbol{b} \in \mathbb{R}^3$

1: **for** $i \in \{1, \ldots, D\}$ **do**                                         ▷ For each unit-types

2:     $x_i \leftarrow \min\limits_{j=1,2,3} \left\lfloor \frac{b_j z_i}{a_{ij}} \right\rfloor$              ▷ Determine the amount of unit to generate.

3:     **for** $j = 1, 2, 3$ **do**                             ▷ For each resource-types

4:         $b_j \leftarrow b_j - x_j a_{ij}$       ▷ Subtract the amount of resources spent to generate units.

5:     **end for**

6: **end for**

7: **return** $\mathbf{x} = \langle x_1, \cdots, x_m \rangle$                       ▷ Return a unit-combination.

---

Table 2: Win-rate and diversity of each agent in the scenario with resources $\boldsymbol{b} = \langle 5000, 2500, 75 \rangle$ and $\boldsymbol{b} = \langle 5000, 4000, 75 \rangle$. DiffEvo($k$) denotes diffusion evolution with temperature hyperparameter $c_T = k$.

| Scenario | Algorithm | Win-rate | | Pairwise Distance | |
|---|---|---|---|---|---|
| | | **Average** | **Best** | **L1** | **L2** |
| | CMA-ES | 0.933602 | 0.989382 | 0.488778 | 0.300154 |
| | DiffEvo (0.1) | 0.751049 | 0.979823 | **0.927559** | **0.505161** |
| $\boldsymbol{b} =$ | DiffEvo (0.03) | 0.938557 | 0.986184 | 0.568563 | 0.313240 |
| $\langle 5000, 2500, 75 \rangle$ | DiffEvo (0.01) | 0.965497 | 0.988985 | 0.392148 | 0.221243 |
| | DiffEvo (0.003) | 0.974009 | **0.989396** | 0.253050 | 0.142561 |
| | DiffEvo (0.001) | **0.974898** | 0.989305 | 0.210469 | 0.116760 |
| | CMA-ES | 0.956598 | 0.990784 | 0.438703 | 0.267986 |
| | DiffEvo (0.1) | 0.740088 | 0.990442 | **0.944798** | **0.513645** |
| $\boldsymbol{b} =$ | DiffEvo (0.03) | 0.936185 | 0.987127 | 0.566562 | 0.311798 |
| $\langle 5000, 4000, 75 \rangle$ | DiffEvo (0.01) | 0.966618 | 0.989794 | 0.389652 | 0.218604 |
| | DiffEvo (0.003) | 0.973329 | 0.990961 | 0.301112 | 0.175171 |
| | DiffEvo (0.001) | **0.975238** | **0.990954** | 0.260404 | 0.152349 |

The surrogate model $\hat{f}$ was implemented as a multi-layer perceptron with architecture MLP($2D, 96, 96, 48, 6, 1$) and trained using cross-entropy loss with a learning rate of $10^{-4}$. The dataset contained $|\mathcal{D}| = 10^5$ samples, with a split ratio of $|\mathcal{D}\text{train}| : |\mathcal{D}\text{validation}| : |\mathcal{D}_{\text{test}}| = 0.64 : 0.16 : 0.20$. For the CMA-ES baseline, we used the recommended hyperparameters from Hansen (2016), with initialization $\boldsymbol{\mu} = \boldsymbol{0}$ and $\sigma = 0.5$.

We assessed solution quality by measuring the average and best win-rates of strategies, and solution diversity by computing the pairwise $L_p$ distance between strategies. Formally, the average and best win-rates are defined as

$$[\text{mean}, \text{max}]_{\boldsymbol{x} \in \boldsymbol{U}_0 \cup \boldsymbol{V}_0} \frac{1}{N_z} \sum_{j=1}^{N_z} \hat{f}(\boldsymbol{x}, \boldsymbol{z}_j; \theta_0), \tag{12}$$

and the pairwise $L_p$ distance as

$$\frac{1}{2N(2N-1)} \sum_{\substack{\boldsymbol{x}_1, \boldsymbol{x}_2 \in \boldsymbol{U}_0 \cup \boldsymbol{V}_0 \\ \boldsymbol{x}_1 \neq \boldsymbol{x}_2}} ||\boldsymbol{x}_1 - \boldsymbol{x}_2||_p, \tag{13}$$

where $\boldsymbol{U}_0 \cup \boldsymbol{V}_0 = \{\boldsymbol{u}_0^{(i)}, \boldsymbol{v}_0^{(i)} : i \in \{1, \ldots, N\}\}$ and $\boldsymbol{z}_j \sim_{iid}$ Dirichlet($\boldsymbol{1}$), $N_z = 128$. The results, summarized in Table 2, indicate a trade-off between diversity and performance. Specifically, as the temperature parameter increases ($c_T = 0.01$ or $0.03$), population diversity increases while win-rates decrease. Conversely, as the temperature decreases ($c_T = 0.003$ or $0.001$), diversity diminishes while win-rates improve.

## 5 CONCLUSION

This paper introduced Diffusion Co-evolution to address the challenges of learning stability and strategic diversity in multi-agent environments like StarCraft II. We demonstrated that our approach effectively discovers multiple high-performing strategies and maintains diversity, outperforming conventional baselines such as CMA-ES in both stability and robustness. These results highlight the potential of diffusion-based evolutionary methods in complex, competitive domains, with a scalable and efficient multi-agent strategy learning. Future work could explore applying diffusion co-evolution to larger-scale environments, incorporating richer agent behaviors such as micro-management, and extending the method beyond StarCraft II to other domains where robust multi-strategy learning is essential.

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
