# OpenReview forum: "Stable and Diverse Strategy Learning via Diffusion-Based Co-Evolution in StarCraft II Combat"
_ICLR.cc/2026/Conference — ICLR 2026 Conference Withdrawn Submission_

### Official Review · Reviewer_M9L2 · 2025-10-25

**Soundness:** 3
**Presentation:** 2
**Contribution:** 2
**Rating:** 2
**Confidence:** 3

**Summary:**

This paper incorporates the idea of using diffusion into multi-agent competitive games. The approach leverages on the diffusion algorithms capability to generate diverse set of agents to approach the Nash equilibrium. The algorihm was experimented on Starcraft II benchmark and showed promising results against CMA-ES

**Strengths:**

Regarding the mixed-policy Nash equilibrium, using diffusion is a reasonable choice in solving the multiagent problem. The justification for using diffusion was well explained, and the results from the Starcraft benchmark is interesting.

**Weaknesses:**

First, the algorithm is not explain well. Several notation, including major notations such as 'u,v,y' is not made clear. It is therefore difficult to measure the novelty nor the significant of the algorithm due to lack of clarification on the algorithm.

Several experiment details are missing. While the hyperparameters are explained well, the other details, such as how the policy is represented as, is not explained well. Also, it would be nice to give a quick summary on the types of missions included in the Starcraft Benchmark, including the qualitative and quantitative summary on the tasks involved.

While the idea is interesting, just comparing the algorithm with CMA-ES alone doesn't seems to be enough, as just one baseline isn't enough for a comprehensive comparison. Places to start for baselines would be Quality-Diversity papers. As diffusion in the context of this paper is used to generate diverse set of high-performing policies, Quality Diversity algorithms would highlight whether the diffusion algorithm indeed excels by giving diverse set of policies to choose from.

Finally, some of the writing style can be improved. For example, Figure could use a temperature scale bar to show exactly which colors corresponds to which values.

**Questions:**

Other than clarifications requested above, I think it would be interesting to see a more comprehensive experiments, with more baselines. It would perhaps be better to see experiments with benchmarks other than Starcraft II. This would show how the proposed algorithm scales to other types of benchmarks as well.

---

### Official Review · Reviewer_QnvB · 2025-10-27

**Soundness:** 2
**Presentation:** 2
**Contribution:** 2
**Rating:** 2
**Confidence:** 4

**Summary:**

The paper proposes a Diffusion Co-evolution framework to learn stable and diverse strategies in a StarCraft II (SC2) “combat” setting. It borrows a diffusion-evolution update (Equations (1)–(4)), learns a surrogate simulator from battle rollouts, then runs a two-population diffusion procedure with a Boltzmann reweighting and a cosine schedule to update strategy populations U,V (Alg.1). Evaluation reports win-rate and pairwise distances as “diversity”, and shows temperature–performance trade-offs in two hand-picked resource scenarios (Table 2). Claims: better stability and more diverse high-performing strategies than CMA-ES in SC2 macro unit-composition planning.

**Strengths:**

1. Clear, reproducible description of the surrogate-then-evolve pipeline and diffusion updates (Alg.1; Eqs.1–4).

2. The paper does surface the well-known cyclic dominance issue in competitive domains and tries to maintain multiple strong modes, rather than a single point solution (Fig.2).

3. Ablating temperature shows a tangible performance–diversity trade-off (Table 2), which is at least honest about the dial you are really turning.

**Weaknesses:**

1. “Diffusion models are evolutionary algorithms” and Diffusion Evolution per se are established; this paper mostly slots them into a co-evolution loop with a temperature control and a simple surrogate. The core idea (use a diffusion-like denoising step as an ES to preserve multiple optima) is already articulated by Zhang et al. (ICLR’25) with stronger analyses and broader baselines.(https://openreview.net/forum?id=xVefsBbG2O) Here, there is no rigorous connection to PSRO / Nash-seeking procedures widely used for cyclic, non-transitive games (Lanctot et al.’17; surveys and fast EPSRO variants). (https://openreview.net/pdf/f5f84bde2c5f67e595132f7d301d41e2d57d7026.pdf) and (https://www.ijcai.org/proceedings/2024/0880.pdf) Simply evolving two populations without meta-game reasoning is not new in SC2; league training / exploiters / PSRO are the de-facto baselines.(https://deepmind.google/discover/blog/alphastar-grandmaster-level-in-starcraft-ii-using-multi-agent-reinforcement-learning/)

2. a) Reporting best/mean win-rate against random Dirichlet-sampled opponents (Eq.12) and distances as “diversity” (Eq.13) says nothing about exploitability, Nash mixing, or population robustness. Competitive evaluation needs Nash averaging / meta-game analysis, exploitability, or at least response-to-a-league (Balduzzi et al.’18; PSRO literature).(https://arxiv.org/abs/1806.02643) and (https://proceedings.neurips.cc/paper_files/paper/2018/file/cdf1035c34ec380218a8cc9a43d438f9-Paper.pdf)

2. b) The paper never computes exploitability or a Nash-averaged score. With cyclic rock–paper–scissors structure (Fig.2), your “best strategy” metric is meaningless.

2. c) No league-style baselines (AlphaStar framework; robust opponent selection; goal-conditioned exploiters), despite SC2 being the canonical domain. (https://deepmind.google/discover/blog/alphastar-grandmaster-level-in-starcraft-ii-using-multi-agent-reinforcement-learning/)




3. The entire algorithm optimizes on (Alg.1 lines 16–31), yet we see no calibration curves, OOD stress, or human-grounded error bars A single MLP trained with CE on a synthetic dataset of size 10^5 (80/20 split) is extremely fragile for SC2 combat dynamics. What happens when f is mis-calibrated in a cyclic region? You may be diffusing towards a simulator artifact.

4. You drop micro-management entirely and reduce SC2 to a static resource allocator (Sec.2.2; Alg.3). That is acceptable as a toy, but then comparing to CMA-ES only on two hand-picked budgets is not a result for ICLR. Modern SC2 works test league growth, robustness, offline datasets, PSRO-style meta-game—none of which appear here.(https://www.researchgate.net/publication/372962307_AlphaStar_Unplugged_Large-Scale_Offline_Reinforcement_Learning)

5. Using pairwise L_1/L_2 distances in the allocation vector (Table 2) is behavior-agnostic. QD literature emphasizes behavior descriptors (e.g., army time-to-kill profiles, matchup-specific outcome vectors, micro-robustness) and MAP-Elites/QD grids for maintaining truly functionally diverse elites (Mouret & Clune’15; QD surveys). (https://members.loria.fr/jbmouret/qd.html) and (https://quality-diversity.github.io/papers.html) and (https://dl.acm.org/doi/10.1145/3319619.3321904) A cosine in parameter space is not “strategy diversity”.

6. A few curves with different c_T show a performance–diversity trade-off (Table 2), but there’s no seed variance, no training-time oscillation analysis, no Red-Queen/cycling diagnostics, no PRD/DOM population metrics. In competitive learning, “stable” means resistant to cycles under adversarial best responses; that is never tested.

7. If the goal is “diverse, high-performing strategies in cyclic games,” the minimal baselines are: PSRO / EPSRO, Nash averaging, league training with exploiters, and QD (MAP-Elites) with behavior descriptors. None are present.(https://proceedings.neurips.cc/paper_files/paper/2023/file/94796017d01c5a171bdac520c199d9ed-Paper-Conference.pdf)

8. Eqs.(1)–(4) are standard diffusion-evolution denoising with schedule \alpha_t, noise \sigma_t, and a reweight g. There is no theorem or stability analysis in the co-evolution setting (two interacting pops with cyclic payoffs). You owe at least a fixed-point or contraction discussion or meta-game view.

**Questions:**

1. Report Nash averaging (Balduzzi’18), exploitability, and meta-game response graphs. Otherwise your results are not meaningful for cyclic SC2.

2. You claim robustness and diversity in a non-transitive game; PSRO family is the baseline. Add PSRO/EPSRO and a league with goal-conditioned exploiters (Huang et al., NeurIPS’23).

3. Provide calibration, error heatmaps over the strategy-strategy grid, OOD tests, and end-to-end re-evaluation of evolved populations in the true simulator (not \hat f). Quantify the sim-to-surrogate gap per scenario.

4. Replace L_p in allocation space with QD-style behavior descriptors and MAP-Elites coverage/PRD. Show that your “diverse” set spans functionally distinct strategies, not just different vectors.

5. Is your diversity just temperature? Try alternative re-weights, schedules, and show sensitivity across seeds.

6. Drop your evolved populations into a PSRO/league and show how quickly they are exploited. If quickly, your “stability” claim collapses.

7. Add micro or at least perturbation robustness (fog of war, noisy micro skill). Right now this is a static allocator with a convenient surrogate—far from SC2.

8. Provide any convergence/stability analysis for the two-population diffusion update under non-transitive payoffs, or at least a meta-game perspective.

---

### Official Review · Reviewer_xHfi · 2025-11-01

**Soundness:** 2
**Presentation:** 2
**Contribution:** 1
**Rating:** 2
**Confidence:** 3

**Summary:**

The paper proposes Diffusion Co-evolution, an evolutionary framework for competitive multi-agent settings (specifically StarCraft II unit-composition combat) that borrows the sampling-and-noise schedule of diffusion models to keep a population of strategies both performant and diverse. On two StarCraft II resource settings that the authors identify as having such cycles, the method produces populations with win rates comparable to CMA-ES, while achieving higher pairwise diversity for appropriate temperature choices.

**Strengths:**

- The paper is clear about the research gap. The single-point convergence undesirable under competitive settings due to cyclical nature of the optimization landscape. The paper frames the goal as finding a diverse set of strong strategies rather than a single best strategy.

**Weaknesses:**

- Evaluation scope is narrow. All the results are in a highly simplified SC2 setting: it is essentially a resource-to-unit-composition mapping with fixed micromanagement, i.e. a low-dimensional continuous allocation game, which does not include unit micromanagement.
- Baselines are sparse. The only baseline is CMA-ES . There is no comparison to established game-population / meta-game approaches such as PSRO / NFSP / other quality-diversity co-evolution (MAP-Elites variants). Without those, it is impossible to tell whether the benefit comes from “using diffusion” or simply from “using any population method with explicit diversity pressure.” This is probably the biggest blocker.
- The results are not analyzed statistically. It's hard to tell the significant of the improvements.
- The surrogate model is a crucial component but not analyzed. The method leans heavily on a learned MLP simulator f̂ trained on 10⁵ samples, and then all subsequent co-evolution is done against that surrogate. But we never see (i) calibration of f̂ on out-of-distribution compositions generated later by diffusion, (ii) how sensitive the final diversity is to surrogate noise, or (iii) how well this surrogate work under more complex environments.

**Questions:**

- The paper assumes fixed micro-management so the game reduces to allocation. How would this plug into a hierarchical SC2 agent where the upper level generates unit compositions and the lower level is an RL policy? Please comment on compute and surrogate feasibility.

---

### Official Review · Reviewer_bhyV · 2025-11-03

**Soundness:** 1
**Presentation:** 1
**Contribution:** 1
**Rating:** 0
**Confidence:** 3

**Summary:**

The paper uses Diffusion Evolution to train a distribution of SCII agents with diversity, with the fitness function defined as the average win-rate. The proposed method focuses on SCII macro management - specifically resource allocations.

**Strengths:**

The idea of using Diffusion Evolution for Quality-Diversity optimization could be interesting, but the paper needs major rework.

**Weaknesses:**

The overall writing of this paper is confusing and incomplete. In each section the authors use a completely different math language. There are no consistencies across sections.

Section 2 does not give a self-contained description of Diffusion Evolution, which is a fundamental piece of the proposed method. Section 2.2 lacks clarity as a lot of the variables are not clearly defined - more in the Questions section.

The method section (Section 3) is more of a high-level introduction with important details missing - For example, what is the purpose of the surrogate simulator and how is it trained? The demonstrative figure (Figure 1) is hard to interpret and has no caption. Same for Algorithm 1, most of the variables in it are either not defined or have a different formulation. The comments are also not informative.

I'm also not sure what the takeaway is from the experiments. It seems that the goal is to establish a Pareto front with a quality-diversity trade-off, but there is no such figure. From Table 2, the only observation is that a DiffEvo agent population either has a lower average win-rate or a lower diversity compared with the only baseline CMA-ES.

**Questions:**

- Eq 5: Define $S$, $\Theta$. Writing $f_{agent}(\cdot)$ equaling a constant $p$ is incorrect.
- Eq 6, 7: Give more context on $x_1, x_2$ and $\mathcal{X}_1$, $\mathcal{X}_2$. Are they scalars? Vectors? Integers? Are they the resource allocation of the two players? What is $p(x_2)$? Where is this distribution taken from?
- Figure 2: What is the need to show a 40x40 win-rate heatmap if the point is simply that cyclic behaviors exist? Why not just show the win-rate table of the cycle?
- Algorithm 3 is referenced before Algorithm 2.

---

### Note · Authors · 2025-12-03

**Comment:**

Thank you for taking the time to review our manuscript and for sharing your valuable insights. After discussing the constructive feedback you provided, we have decided to withdraw the paper. We were fortunate to receive thoughtful comments from the four reviewers with expertise in multi-agent systems and quality diversity, and we are grateful for the depth of insight you offered.

While the reviewers agreed that the manuscript addresses an important problem, their comments highlighted several key areas requiring substantial improvement. In retrospect, the academic positioning of the paper was not articulated clearly, which ultimately affected the coherence and direction of the work. We also recognized that inconsistencies in the mathematical notation made the paper difficult to follow, a challenge further compounded by the insufficient background explanation of the multi-agent component.

To establish a clearer direction, restructure the manuscript accordingly, and present the experiments more effectively, we have chosen to withdraw the current version. We plan to thoroughly revise the manuscript based on the valuable feedback you provided and submit an improved version in the future. Thank you again for your thoughtful review.

**Withdrawal Confirmation:**

I have read and agree with the venue's withdrawal policy on behalf of myself and my co-authors.